# Local and global feature aggregation for accurate epithelial cell classification using graph attention mechanisms in histopathology images

**Ana Leni Frei**[1]                                                                        ANA.FREI@UNIBE.CH
**Amjad Khan**[1]                                                                        AMJAD.KHAN@UNIBE.CH
**Linda Studer**[1,2,3]                                                                LINDA.STUDER@UNIFR.CH
**Philipp Zens**[1]                                                                    PHILIPP.ZENS@UNIBE.CH
**Alessandro Lugli**[1]                                                        ALESSANDRO.LUGLI@UNIBE.CH
**Andreas Fischer**[2,3]                                                    ANDREAS.FISCHER@HEFR.CH
**Inti Zlobec**[1]                                                                        INTI.ZLOBEC@UNIBE.CH

[1] *Institute of Tissue Medicine and Pathology, University of Bern, Switzerland*

[2] *Document, Image and Video Analysis Research Group, University of Fribourg, Switzerland*

[3] *iCoSyS, University of Applied Sciences and Arts Western Switzerland, Fribourg, Switzerland*

**Editors:** Under Review for MIDL 2023

## Abstract

In digital pathology, cell-level tissue analyses are widely used to better understand tissue composition and structure. Publicly available datasets and models for cell detection and classification in colorectal cancer exist but lack the differentiation of normal and malignant epithelial cells that are important to perform prior to any downstream cell-based analysis. This classification task is particularly difficult due to the high intra-class variability of neoplastic cells. To tackle this, we present here a new method that uses graph-based node classification to take advantage of both local cell features and global tissue architecture to perform accurate epithelial cell classification. The proposed method demonstrated excellent performance on F1 score (PanNuke: 1.0, TCGA: 0.98) and performed significantly better than conventional computer vision methods (PanNuke: 0.99, TCGA: 0.92).

**Keywords:** digital pathology, malignant epithelial cells, cell classification, cell-based graphs, node classification, graph attention, graph clustering

## 1. Introduction

In the field of digital pathology, recent advances in deep learning have led to the development of cell-level tissue analyses from hematoxylin and eosin (H&E) stained slides, opening the possibility to conduct accurate quantitative analyses on whole slide images (WSI) that might not be possible in a clinical scenario with a microscopic assessment of slides. In the era of personalized medicine, understanding the precise composition and the spatial histologic features of the tissues is a key point to improve our understanding of tumor behavior (Baxi et al., 2022). Automated cell detection and classification on H&E WSI can be used to explore specific cell-cell interactions and global tissue structure and organization (Ahmedt-Aristizabal et al., 2022; Pati et al., 2022).

Although publicly available models and datasets for cell classification in colorectal cancer exist, these often lack the differentiation between normal and malignant epithelial cells, as

they are usually grouped together as a single "epithelial" class (Graham et al., 2019, 2021). For downstream cell-based analyses, however, the distinction between these two epithelial categories is of great importance. This task is particularly challenging because of the high morphological heterogeneity of neoplastic epithelial cells. Nevertheless, when looking at the overall epithelial gland architecture, the differentiation between normal and malignant glands is easier to make. Taking advantage of this, we propose a new method based on the aggregation of local cell morphology features together with surrounding gland structure information to learn accurate epithelial-cell level classification from H&E images.

## 2. Material and Methods

A subset of patches from the Lizard dataset (Graham et al., 2021) was selected that solely contain either normal or malignant epithelial cells. Additional patches (1000×1000px) from scanned colorectal tissue slides from the Institute's cohort and from TCGA were retrieved and annotated for epithelial cells by experts. The TCGA and PanNuke (a subset of Lizard) patches were kept as test data while the remaining patches were used for training and validation. All patches were extracted at 20X magnification (0.5$\mu$m/pixel). The dataset description can be found in Table 1.

First, epithelial cell tiles (128×128px) were extracted around the cell centroids. ResNet18 (He et al., 2016) and ViT16 (Dosovitskiy et al., 2021) were trained for normal versus malignant cell classification as baseline. ResNet18 showed a considerably better performance than ViT16 on TCGA, and was thus selected as node feature extractor when building the graphs (see Table 2).

Epithelial cell graphs were built on the training patches. Nodes were individual epithelial cells. Node features were extracted from the last hidden layer of the previously trained ResNet18 to describe the cell morphology. Edges were built using Delaunay triangulation (Delaunay, 1934) to connect the nodes (epithelial cells). The best performing Graph Neural Network (GNN) architecture was obtained by testing and optimizing the following parameters using a 5-fold cross-validation: the type of Message Passing function (MP) [GCN, GraphSage, GIN, GATv2] (Kipf and Welling, 2017; Hamilton et al., 2018; Xu et al., 2019; Brody et al., 2022), the number of MP layers [2, 3, 4] and the size of the MP layers [64, 128, 256, 512]. The edge length threshold was also optimized.

Following the graph-based classification, some single misclassified nodes remained, Figure 1 step 4. Thus, a final post-processing (pp) step was applied to smooth the graph predictions. As individual epithelial glands are expected to be composed of either normal or malignant epithelial cells, single glands were isolated into subgraphs using a short edge

| datasets | epithelial type | #patches | #cells | average #cells/patch |
|---|---|---|---|---|
| **train data** | normal | 68 | 66,034 | 797 |
| **(Lizard + Institute)** | malignant | 107 | 119,013 | 1093 |
| | normal | 14 | 9,301 | 664 |
| **test data** | malignant | 17 | 12,335 | 726 |
| **(PanNuke + TCGA)** | both | 5 | normal: 3,450 | 690 |
| | | | malignant: 5,362 | 1,072 |

Table 1: Overview of the number of patches and cell types in the different subsets.

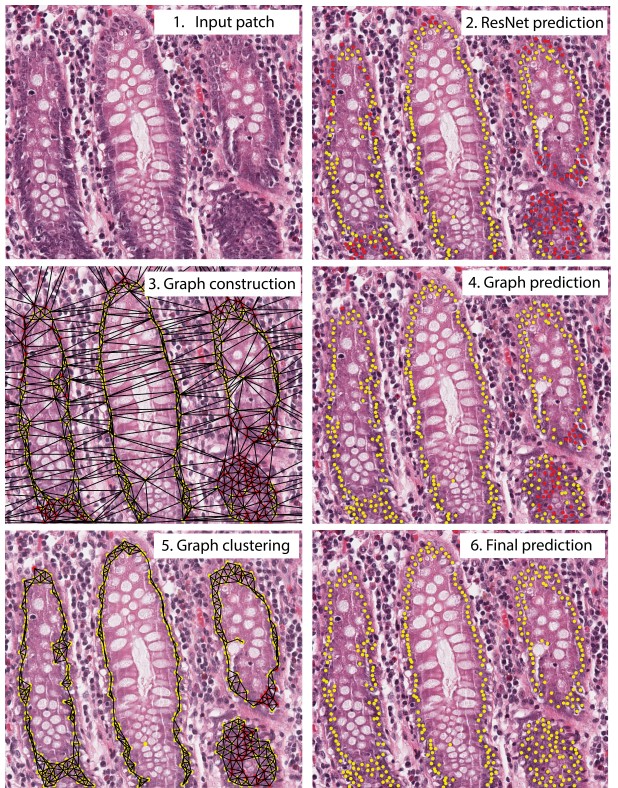

Figure 1: Epithelial cell classification pipeline on a normal epithelial tissue example from TCGA testset. Red dots indicate epithelial cells classified as malignant and yellow dots cells classified as normal. Black lines indicate edges between connected cells when applying graph classification and graph clustering.

Table 2: Weighted F1 score.

| Model | PanNuke | TCGA |
|---|---|---|
| ViT16 | 99.099 | 89.469 |
| ResNet18 | 99.039 | 91.716 |
| GAT | 99.423 | 95.074 |
| GAT + pp | **100.0** | **97.847** |

threshold of 30px(15$\mu$m). For every single epithelial gland, a median filter was applied to all cells in that gland to get the final cell class. Each step is illustrated in Figure 1.

## 3. Results

Normal cells were frequently misclassified as malignant in denser cell regions (such as epithelial crypt base) by standard CNN methods. Using graph-based cell classification solved this issue and significantly improved the cell classification ($p < 0.05$) as can be seen in Table 2. The best-performing method was a GNN composed of 4 GATv2 layers of size 256. The final clustering and filtering further improved the cell classification.

## 4. Discussion and Conclusion

We show that graphs allow us to effectively capture the structural context around the cell of interest for accurate epithelial cell classification in colorectal H&E images. The proposed graph-based model is highly accurate and can be easily applied in addition to any other model detecting epithelial cells to further differentiate between normal and malignant cells.

## Acknowledgments

This work was funded by the Swiss National Science Foundation (CRSII5_193832). Results presented here are based on data provided by the TCGA Research Network.

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
