# OpenReview forum: "Local and global feature aggregation for accurate epithelial cell classification using graph attention mechanisms in histopathology images"
_MIDL.io/2023/Short_Paper_Track — MIDL 2023 Short paper track Poster_

### Official Review · Reviewer_o2P9 · 2023-04-21
**The method is novel, the view is clear, the structure is proper, the experimental treatment and results can be more detailed**

**Rating:** 7
**Confidence:** 4

**Review:**

The significance of this work:
The authors describes a new method for accurate epithelial cell classification in digital pathology, which outperforms conventional computer vision methods. This method uses graph-based node classification, graph attention, and graph clustering to take advantage of both local cell features and global tissue architecture. The proposed method can differentiate between normal and malignant epithelial cells, which is important for downstream cell-based analysis. This new method has the potential to improve cell-level tissue analysis and provide more accurate information about tissue composition and structure.In my opinion,the idea is interesting and the authors have performed some experiments to show the effectiveness of the proposed method. Generally, the paper is well written. But I still have some concerns that are expected to be addressed in a revision:
(1)The article subtitle "Materials and methods" " is mentioned in the third paragraph， "exploring the following parameters: number of message passing (MP) layers [2, 3,4], size of the MP layers [64, 128, 256, 512] for the most common MP functions [GCN, GraphSage, GIN, GATv2] ",Whether the authors derive the best configuration from the experiments.
(2)How does the author say that "because ResNet18 performs better than ViT 16 on TCGA data, but the performance is similar on PanNuke"?
(3)It seems that the author did ViT related experiments, but why did not see the results of the visualization? I think the description of the experimental results can be further optimized for the convenience of the reader.

---

### Official Review · Reviewer_gevF · 2023-04-26
**Review of Local and global feature aggregation for accurate epithelial cell classification using graph attention mechanisms in histopathology images**

**Rating:** 8
**Confidence:** 3

**Review:**

This paper presents a new method for classifying epithelial cells from hematoxylin and eosin (H&E) stained histopathology images. The strategy is to use graph neural networks (GNN) for global feature aggregation combined with a ResNet for local features.

* One point that is unclear in this paper is that the goal of the graph seems to basically be to get a consistent classification for all cells in an epithelial gland (i.e., they should all be normal or all malignant). If this is indeed the motivation, it should be made more clear in the introduction.

* The classification rates are already quite high for the ResNet alone, which is classifying each cell independently of the others and without more global image context. The F1 score for PanNuke was already 0.99 and TCGA was 0.917. Is ResNet on individual epithelial cells really the state-of-the-art? Is there not another baseline method that also works with more global context?

* The only images shown are of a normal image where everything worked perfectly. It would be informative to see a more difficult case and maybe even a case where there are some misclassifications. (I know this is a short paper, so space is limited.)

Overall, this is an interesting idea to use graph neural networks to incorporate global context into epithelial cell classification and worthy of publication in MIDL.